# The Poetry of Recovery in Peer Support Workers with Mental Illness: An Interpretative Phenomenological Analysis

**DOI:** 10.3390/healthcare12020123

**Published:** 2024-01-05

**Authors:** Su-Yeon Kim, Young-Ran Kweon

**Affiliations:** 1Department of Nursing, Honam University, Gwangju 62399, Republic of Korea; 2022091@honam.ac.kr; 2Department of Nursing, Chonnam National University, Gwangju 61469, Republic of Korea

**Keywords:** mental illness, peer support, recovery, experience, interpretative phenomenological analysis

## Abstract

This study was conducted to investigate and understand various aspects related to participants’ experiences in peer support activities, with a particular focus on their personal growth and the influence of these activities on their lives. In this qualitative study, peer support workers with mental illness were the main subjects, and they were recruited from G Metropolitan City, South Korea. The study used purposive sampling, guided by recommendations from peer support worker support organizations. A total of five participants were selected using purposive sampling to ensure maximum variability in the sample. Data collection involved semi-structured individual interviews, and data analysis was conducted using the interpretative phenomenological analysis (IPA) method. Following the IPA procedure for data analysis, the study revealed six themes that encapsulated the recovery experiences of peer support workers with mental illness: (1) Facing confusion and challenges, (2) Rising and refining myself, (3) Navigating the paths of relationships, (4) Gazing at the desired horizons, (5) Awakening the inner hero, and (6) Standing as a person who cherishes life. This research underscores the positive impact of peer support activities on individuals who have faced mental health challenges. It emphasizes the significance of self-discovery, the development of supportive relationships, and the aspiration for a brighter future. These findings contribute to the expanding body of knowledge regarding the benefits of peer support in the context of mental health recovery.

## 1. Introduction

The recent significant shift in mental health promotion programs emphasizes the rehabilitation, well-being, and protection of rights for individuals with mental illnesses [1]. This shift aligns with the recovery paradigm in mental health, which focuses on individuals with mental health challenges actively participating in their local communities, working, learning, and discovering the meaning and purpose of their lives [2]. In line with this, recovery-oriented practice (ROP) has emerged as a pivotal framework, advocating for approaches that are person-centered and strength-based, recognizing the unique journey of each individual toward recovery [3]. ROP stresses the importance of empowering service users, advocating for their active involvement in decision-making processes, and fostering environments that support personal growth and recovery [4].

The recovery paradigm within the mental health field, particularly under the ROP framework, significantly emphasizes the role of peer support activities [5]. This paradigm advocates for individuals with mental illnesses to be active participants in their recovery process, rather than passive recipients of care dictated by treatment providers. Research has consistently shown that crucial elements aiding individuals with mental illnesses in shaping their lives include their relationships with mental health professionals, family members, and peers who have had similar experiences [6,7,8,9]. The importance of peer support is further underscored by scholars who have highlighted its role in facilitating learning and personal growth for those facing mental health challenges [10,11]. This aligns with a service user-driven care model, which focuses on maximizing self-determination and leveraging individual strengths and recovery potentials [5,12].

In the realm of mental health, the role of a ‘peer support worker’ is defined as individuals providing support through their own experiences with mental health issues [13]. Research on peer support in mental health has significantly expanded since 2010. Earlier studies primarily focused on the experiences of those receiving peer support and the effectiveness of such services [14,15,16,17]. Yet, a considerable research gap exists in understanding the dual role of peer support workers [18,19]. These individuals play a unique role by not only assisting others in their recovery journey but also managing their own mental health challenges. This dual responsibility is particularly pertinent within the ROP framework. In contexts like South Korea, it is crucial to understand and improve the effectiveness of peer support from the viewpoint of these providers. Investigating the experiences of individuals who have faced mental illnesses themselves and subsequently become peer support workers is vital. Their unique insights and deep understanding of the recovery process’s complexities and nuances are invaluable.

Qualitative research is the most appropriate methodology for exploring these complex, multi-dimensional experiences, as it allows for an in-depth examination of the subjective, personal, and contextual dimensions of recovery [20]. Phenomenological research, deeply rooted in the tangible and vivid essence of human experiences and the subjects’ viewpoints, aims for an intuitive understanding of the meaning of experiences and the fundamental aspects of existence “as they are” [21]. Complementing this philosophical foundation, interpretative phenomenological analysis (IPA) offers a well-defined methodology, including hermeneutic cycles and idiographic techniques, to enhance the analysis and enable a deeper exploration of the meaning inherent in phenomena [22]. Essentially, while phenomenological research seeks to grasp human experiences in their purest form, IPA provides a structured and systematic approach to interpretation, with a specific focus on extracting meaning from phenomena [23].

IPA, as a qualitative research method, embraces a double hermeneutic process [24]. This process extends beyond merely interpreting raw data to encompass the interpretations and understandings of the participants themselves. This approach, which ensures a comprehensive understanding that is firmly grounded in the participants’ unique perspectives [25], is particularly valuable in mental health research. By being rooted in empathetic understanding and real life experiences, it not only fosters a deeper connection with the subject matter but also promises to yield insights that are both profound and directly applicable to patient-centered interventions. When applied to individuals who have experienced mental illnesses and have transitioned into the role of peer support workers, IPA retains the authenticity of their voices while integrating layers of interpretation to reveal profound insights.

Therefore, exploring the experiences of peer support workers becomes particularly crucial within the ROP framework, as it aligns with the overarching goal of promoting person-centered care, fostering inclusivity, and embracing empathetic approaches in mental health services. This study aims to contribute not only to the advancement of recovery-oriented peer support activities but also to laying the groundwork for the establishment of the role of community mental health nursing. The findings will serve as foundational material for informing the development of nursing interventions that align with the principles of ROP, advocating for holistic and person-centered care in the realm of mental health services.

## 2. Materials and Methods

### 2.1. Study Design

This study employed a qualitative research approach to deeply explore the lived experiences of peer support workers, with a specific focus on applying the interpretative phenomenological analysis (IPA) method [24]. IPA is particularly well-suited for this research as it facilitates an in-depth exploration into how individuals make sense of their personal and professional experiences. This methodological choice aligns closely with our aim to understand the nuanced dynamics of recovery and support in the context of mental illness. By focusing on the subjective meaning-making process, IPA enables a detailed investigation into the perspectives and realities of these individuals. The exploratory and descriptive nature of this research design not only allowed for a comprehensive examination of the lived experiences of peer support workers but also provided valuable insights into their unique journeys of recovery and the impact of their role in the mental health support system.

### 2.2. Participants

The participants in this study were selected using purposive sampling, guided by recommendations from peer support worker support organizations in G Metropolitan City, South Korea. The selection criteria included individuals with mental illnesses, such as schizophrenia and bipolar disorder, who were actively undergoing treatment and managing their conditions with consistent medication. They had also completed peer support worker training and had been engaged in peer support activities for over a year. Individuals with impaired cognitive abilities that hindered effective communication or those who were unable to sufficiently express their experiences were excluded from the study.

Regarding the study’s sample size, adherence to the methodological norms for IPA was crucial. According to Smith [25], a prominent IPA scholar, the ideal sample size usually lies between three and six participants. This size is considered sufficient to reveal significant patterns and variances in the experiences of participants while avoiding the risk of data saturation. Smith emphasizes in his seminal works the risks of a too-large sample size in IPA studies, which can dilute the depth of individual experiential narratives, thereby potentially undermining the study’s validity [25]. Considering these factors, the study used a sample size of five. This choice, guided by the mentioned methodological principles, aimed to balance thorough, individualized examination of each participant’s experience with a manageable scope of analysis. Participants were chosen via purposive sampling, with nominations from organizations involved in peer support initiatives. This selection strategy was key to ensuring a sample representative of the target demographic, thus enhancing the study’s relevance and the potential application of its findings. Therefore, a sample size of five was considered optimal for attaining the depth and detail integral to IPA research.

The general characteristics of the study participants are shown in Table 1. The participants included 1 male and 4 females, with ages ranging from 40 to 59 years. Marital status consisted of 3 married and 2 divorced individuals. Two participants were receiving regular salaries through peer support worker job programs, while the other 3 received irregular payments for their peer support activities. Educational backgrounds varied, with 2 participants having completed high school, 1 with a diploma from a vocational college, 1 who withdrew from a university, and 1 with a university degree. Diagnoses included 2 individuals with bipolar disorder and 3 with schizophrenia. The duration of illness ranged from 13 to 27 years, with 3 of them registered as disabled and 2 not registered.

### 2.3. Data Collection

Data collection occurred between 4 January and 30 April 2021, encompassing one-on-one, in-depth interviews with participants selected purposefully based on recommendations from organizations supporting peer support activities in G Metropolitan City, South Korea. This methodology adhered to the guidelines presented in the second edition of IPA by Smith and Fieldsend [26].

Before initiating data collection, a preliminary review was undertaken to refine the interview questions. The feedback of a mental health nurse, a professor of mental health nursing, and an individual with a mental health condition was solicited. Following their input, the interview outline was adjusted and subsequently employed during data collection.

The initial interview began with a core question: “Could you describe what being a peer support worker means to you?” This question aimed to elicit a broad understanding of the role and significance of being a peer support worker from the participants’ perspectives. To gain deeper insights, subsequent questions encouraged participants to provide comprehensive explanations and elaborate on their emotional experiences associated with this role. Follow-up questions were carefully tailored to each participant’s narrative, ensuring a personalized and relevant line of inquiry. This approach allowed the exploration of individual experiences in greater depth, facilitating a richer understanding of the role and impact of being a peer support worker. Additionally, to enrich the data and provide a more nuanced understanding, supplementary materials such as presentations, notes, and activity diaries provided by the participants were taken into consideration. These materials offered additional context and depth to the participants’ verbal narratives, enhancing the overall comprehension of their experiences. A key additional question posed to the participants was: “How do individuals with mental illnesses, who also serve as peer supporters, understand the influence of peer support activities on their recovery?” This question was pivotal in understanding the dual role of the participants as both providers and receivers of peer support. It aimed to uncover how their experiences as peer support workers influenced their recovery journeys, offering valuable insights into the reciprocal nature of peer support in the context of mental health and recovery.

Each case consisted of 3 to 4 interview sessions, each spanning 45 to 60 min. The primary objectives of the initial and second interviews were to grasp the specific meanings of key questions. The third and fourth interviews were aimed at validating participants’ perspectives on themes and comments. When necessary, the researcher posed questions to delve deeper into subjects or seek clarification.

All interview content was recorded with the participant’s consent. Subsequently, interviews were transcribed after multiple listens to the recorded content. The researcher maintained detailed notes on the interview context, the prevailing atmosphere, and the primary content of each session, and also documented reflective notes regarding empathetic and reflective thought processes.

### 2.4. Data Analysis

After the interviews, the data were analyzed using the IPA, as outlined by Smith and Fieldsend [24], following a six-step process. The details were as follows: (1) Reading and Re-reading. This step included transcribing interviews, reviewing audio recordings, and noting participant characteristics, interview context, and reflections. (2) Initial Noting. During this phase, important sections in transcripts were underlined, exploratory comments were recorded, and observations were categorized as descriptive, linguistic, or conceptual. (3) Generating Initial Themes. The notes from the previous steps were transformed into concise language to create themes that encapsulated the essence of participants’ experiences. (4) Searching for Themes’ Coherence. In this step, the study aimed to identify patterns and relationships among initial themes. Some initial themes were integrated into broader concepts, and higher-level themes were abstracted. The strategies employed for establishing the coherence of these themes were as follows: (a) Abstraction: This involved identifying patterns among the emergent themes to develop an understanding of what could be considered higher-order themes. It included the process of grouping similar themes and creating new, encompassing labels for these clusters. (b) Polarization: The focus here was on exploring the oppositional relationships between themes. Instead of similarities, this approach concentrated on differences, reviewing themes in light of their contrasting aspects. (c) Contextualization: This stage involves understanding the interrelationships among the themes. The aim was to clarify contextual elements or narrative components that connected the themes, thereby adding a layer of depth and coherence to the thematic framework. (5) Moving to the Next Case. The research transitioned from one case to another, approaching each case independently to avoid biases from previous analyses and ensuring an unbiased focus on each case. (6) Capturing Patterns across Cases. In this study, an effort was made to discern relationships and commonalities across cases, grounded in individual characteristics. This was achieved through a rigorous process of multiple reclassifications and reorganizations. Through this meticulous approach, we were able to finalize themes and subthemes, articulated in highly abstracted language. This method allowed for a deeper, more nuanced understanding of the patterns and connections inherent in the data, ensuring a comprehensive and insightful analysis. Examples of the six-step analysis in IPA are presented in Table 2, Table 3 and Table 4.

Additionally, the data analysis was a collaborative process involving multiple research meetings with a mental health nursing professor, a mental health nurse, and the researchers. These meetings provided diverse perspectives, helped organize analysis results and abstract findings, and reached a common consensus. This approach facilitated a comprehensive understanding of the experiences of peer support workers with mental illnesses.

### 2.5. Ethics and Informed Consent

This study was approved by the Institutional Review Board at Chonnam National University (IRB no. 1040198-200917-HR-099-03), adhering to ethical standards following the Declaration of Helsinki. Additionally, the study has been registered with the Clinical Research Information Service, Republic of Korea (CRiS), a Primary Registry of the WHO International Clinical Trials Registry Platform (ICTRP), under registration number KCT0008987 at https://cris.nih.go.kr/cris/index/index.do (accessed on 28 November 2023). Written informed consent was obtained, while ensuring the confidentiality of participants. Before the study, participants received comprehensive explanations regarding its significance, objectives, and procedures. They were informed that interviews would be recorded and reassured about the confidentiality and anonymity of their responses. Participants in this study were compensated for their time and effort with a KRW 30,000 gift voucher, given as a token of appreciation. This compensation was provided unconditionally, ensuring all participants received it regardless of their completion of the interview process. This policy highlights the study’s ethical commitment and respect for participant contributions. Participants were explicitly informed of their right to withdraw from the study at any time and for any reason, and their right to receive treatment, regardless of their participation in the study. Additionally, participants were informed that free counseling services would be available if they experienced psychological distress during the interviews.

## 3. Results

Based on the analysis of data following the IPA procedure, the recovery experiences of peer support workers with mental illness can be classified into 6 major themes and 12 subthemes (see Table 5).

### 3.1. Theme 1: Facing Confusion and Challenges

Participants in this study grappled with confusion and burdens in their roles as peer support workers, which included conflicts with superiors and the need for caution. This often resulted in distress. Their struggle with perfectionism and the desire for excellence led to overwhelming feelings.

#### 3.1.1. Aspiring and Striving Due to a Desire to Excel

Participants felt compelled to acquire extensive knowledge and excel in their peer support worker roles, leading to overwhelming feelings and heightened anxiety. They struggled with perfectionism, faced a learning curve, and experienced trial and error. Participant 2 emphasized the burden of striving for perfection: *“I had a bit of a compulsion to know a lot, but as I continued, I realized that instead of such compulsion, it would be more beneficial for me to accept my current self”.*

#### 3.1.2. Feeling Drained and Encumbered by Symptoms

Participants continued to be affected by their symptoms while engaging in peer support work. They described moments of potential symptom relapse due to the job’s demands, such as crises and increased activities, leading to exhaustion. Participant 4 highlighted increased anxiety and stress triggering their symptoms: *“I keep comparing the present situation to about 15 years ago when I had no symptoms… (omitted) Now, I have to study for school, work as a peer support worker, and maybe that’s why I get stressed and anxious, which triggers my symptoms”.*

### 3.2. Theme 2: Rising and Refining Myself

Engaging in peer support work became a catalyst for personal growth and self-discovery for the participants. They found a path to recovery and rediscovered their true selves, which had previously been concealed due to their illnesses.

#### 3.2.1. Unveiling and Embracing One’s True Self

Engaging in peer support work allowed participants to stop hiding their mental illnesses and share their personal experiences. They stressed the importance of accepting oneself without shame or guilt. Participant 1 emphasized self-acceptance, while Participant 2 encouraged embracing their current selves without denying their problems. According to Participant 1: *“Accepting oneself as they are. Not denying but accepting, taking the medication, adhering to treatment because not denying oneself and accepting is a part of it.”* According to Participant 2: *“…but as I continued this work, I realized that instead of acting as if I have no problems like someone without any, it would be better if I trained to look at my current self as it is”.*

#### 3.2.2. Developing Inner Strength to Overcome Challenges

Participants, through their involvement as peer support workers, discovered untapped potential and broke free from negative self-perceptions caused by their illness. They started recognizing their strengths and latent potential. Participant 3 found self-esteem and a sense of accomplishment through positive feedback, while Participant 5 gained self-confidence and the ability to tackle challenges. According to Participant 3: *“…it’s like even a daughter looks at her father as he is… And then, after participating in the N.J. Hospital support group, I think Mr. J seems to be in the best condition. Because he works, he said things like that, and he received feedback…”.* According to Participant 5: *“…through this peer support activity, I became more self-confident. I can take on any challenge with this newfound self-confidence, and that’s a good thing”.*

### 3.3. Theme 3: Navigating the Paths of Relationships

Participants recognized the hardships their family and loved ones endured due to their mental illnesses. Engaging as peer support workers allowed them to connect with others facing similar challenges, deepening their understanding of life’s meaning and the importance of relationships. They emphasized the value of acknowledgment and support.

#### 3.3.1. Discovering Relationship Significance through Peers’ Experiences

Participants were deeply influenced by the struggles of their peers with similar mental health challenges. Witnessing their peers’ resilience and positive transformations motivated them to make positive changes in their own lives. Participant 3, for example, found inspiration in the confidence and openness of peers in various regions, encouraging her to do the same: *“What appeared different? It was natural for them not to hide any of their conditions, and the process they were going through seemed natural. Then, I watched an interview on YouTube… there was a sense of self-confidence. The interviewee appeared very confident, and I felt that I could do it for others as well”.*

#### 3.3.2. Connecting with Others through Active Listening

Participants realized the importance of establishing connections with others through active listening. They evolved from being individuals in need to becoming capable of aiding others. For example, Participant 2 observed that more people sought her assistance, transforming her role from being someone who needed help to someone who could provide support: *“In interpersonal relationships, there weren’t many people who talked to me about needing help. But now, they want to talk, and it seems like they want to chat, which makes me feel a little better”.*

### 3.4. Theme 4: Gazing at the Desired Horizons

Through peer support work, participants began envisioning a future free from relapse and aimed to live a life similar to those without mental illnesses. They held hopes for an improved mental health environment and a well-established framework for peer support, and they were committed to working tirelessly to make this vision a reality.

#### 3.4.1. Cultivating Hope in Adverse Circumstances

Participants faced prejudice and challenging environments in the context of mental health issues. However, witnessing the achievements of peers who persevered in adverse conditions inspired hope and determination in them. Participant 1, for instance, learned about the CEO of Y Foundation, who overcame significant challenges and achieved success, reinforcing the importance of individual efforts: *“You know Mr. Y, the CEO of Y Foundation, he started with very limited resources, struggled with finances, faced malnutrition, and even had to be hospitalized… they had only one office room… When I visited for the second time after 1 year, they had already occupied the whole floor. Even though it was tough, their relentless efforts bore fruit. That’s how I saw it”.*

#### 3.4.2. Working towards a Vision of the Future and Making Efforts

Participants discussed their aspirations and efforts toward a positive future. They shared their visions for the future and their proactive endeavors to lead active, healthy, and positive lives. Participant 4 expressed a desire to continue peer support work, pursue further education, and engage in various courses and activities to achieve their goals: *“As a peer support worker, I want to continue my activities and also engage in further studies, participate in various courses, and so on”.*

### 3.5. Theme 5: Awakening the Inner Hero

Engaging as peer support workers helped participants discover their inner strength and positive aspects that had been concealed by illness. This process shifted their thinking and behavior, enabling them to regain control of their lives and find purpose and meaning.

#### 3.5.1. Finding the Driving Force and Meaning in One’s Life

Participants found newfound motivation and meaning in their lives, which served as catalysts for personal growth and autonomy. They faced difficulties with resilience and a sense of purpose, awakening their inner hero. For instance, Participant 3 now leads a purposeful and goal-oriented life after engaging in peer support activities: *“Before engaging in peer support activities, it felt like I had no sense of purpose or objectives in life. I was just living day to day without a plan. However, this opportunity enabled me to prepare and set specific goals, and it feels like I have a sense of purpose in my life now”.*

#### 3.5.2. Facing Challenges without Fear

Through peer support activities, participants experienced increased positivity, self-confidence, and pride, which enabled them to embrace new challenges and personal growth. Participant 4 mentioned how newfound self-assurance allowed them to handle an eight-hour job and face challenges with confidence: *“Previously, I couldn’t manage an eight-hour job like this while doing peer support work. I had doubts about whether I could handle it. But now, the self-confidence gained from peer support activities has propelled me to this point. I believe I can do any job now”.*

### 3.6. Theme 6: Standing as a Person Who Cherishes Life

Through peer support activities, participants experienced personal growth, joy, and newfound knowledge. They found stability, independence, and the ability to enjoy basic human rights they had previously been denied. This allowed them to embrace their rights and lead fulfilling lives.

#### 3.6.1. Growing and Evolving without Lingering on Symptoms

Participants found opportunities for learning, personal growth, and financial independence through peer support activities. These opportunities allowed them to fully enjoy their rights and live fulfilling lives. Participant 2 highlighted how economic support and increased income improved their overall quality of life: *“Economically, I received significant support. As I engaged in these activities, my income increased, which enabled me to build more interpersonal relationships, pursue hobbies, and lead a more prosperous life”.*

#### 3.6.2. Advocating for Rights in Safe Spaces

Working as peer support workers provided participants with a sense of security and comfort, akin to working in a haven. This feeling of stability enabled them to contemplate a secure retirement and continue personal growth. Participant 3 emphasized the importance of working with individuals who understand their situation, contributing to their sense of security and stability: *“Working with individuals who are familiar with my situation provides a sense of comfort. It’s as if I can continue to live a stable life in this safety zone until my old age, even with the alarm on, while taking medication. It gives me a sense of stability”.*

## 4. Discussion

The primary objective of this study was to delve into and gain a comprehensive understanding of the various facets associated with participants’ engagement in peer support activities, with a specific emphasis on their personal development and the impact of these activities on their lives. The key points of discussion, based on the six themes identified through this research, are as follows.

### 4.1. Facing Confusion and Challenges

In their roles as peer support workers, study participants encountered a multitude of challenges, leading to a sense of burden and an unwavering pursuit of excellence. The experiences highlighted in Spriggs [27] underscore the importance of overcoming obstacles in the realm of peer support, as it is an integral part of the recovery process. Some participants juggled multiple forms of employment, introducing role conflicts and ambiguity, alongside the challenges stemming from their symptoms. These experiences underscore the complexity of the recovery process and the need for resilience in overcoming hurdles. Peer support activities have a positive impact on mental health service recipients, the peer support workers, and the healthcare system as a whole [12]. Despite these benefits, peer support workers often face challenges and limitations. To successfully establish and maintain these services, collaborative efforts are essential. Continuous support from healthcare professionals, managers, and stakeholders is vital [28]. Additionally, it is important for those considering participation in peer support programs to understand these complex dynamics. The role of support from peers or mentors is especially critical during difficult times, such as when symptoms of mental health issues worsen.

### 4.2. Rising and Refining Myself

Participants, while engaged as peer support workers, embodied the overarching theme of “Rising and Refining Myself”. Many had previously lived in isolation, concealing their mental health challenges and lacking self-awareness. Their involvement in peer support activities empowered them to openly share their experiences and initiate the sharing of personal narratives. This transformation allowed them to unlock their latent potential, akin to discovering hidden treasures within themselves. They evolved into individuals capable of facing challenges and adversity with unique coping strategies, showcasing resilience. Young and Ensing’s research underscores the significance of overcoming obstacles, empowerment, learning, self-redefinition, skill acquisition, and life quality enhancement in the recovery process [29]. This study emphasizes self-acceptance, peer support during difficult moments, and the understanding that recovery is an ongoing process, even when symptoms persist [30]. Furthermore, these participants’ journey towards self-improvement, despite facing challenges, highlights the transformative impact of peer support activities, offering inspiration to others on their recovery path.

### 4.3. Navigating the Paths of Relationships

The participants, engaged as peer support workers, highlighted the central theme of “Navigating the Paths of Relationships”. Individuals dealing with long-term mental health challenges often struggle with limitations in their social skills due to their condition, a critical aspect of the recovery process for those with mental disorders [31,32]. This underscores the importance of fostering healthier relationships through peer support activities and the continuous development of these relationships throughout the recovery journey. Previous studies have shown that organizational culture and relationships with colleagues significantly impact job satisfaction among peer support workers with mental health issues [33]. Another study pinpointed a predominant obstacle encountered by peer support workers, which was a feeling of exclusion stemming from an inadequate understanding of the peer support role in the workplace [34]. Recognizing that these limitations in social interactions can significantly affect overall well-being and recovery, mental health professionals should acknowledge the pivotal roles played by optimism, emotional well-being, and social connections in the recovery process. Moreover, the transformative relationships experienced within families and peer groups, as emphasized by the participants, underscore the importance of providing support and guidance in establishing and nurturing positive relationships as a crucial part of the recovery journey [35]. Mental health interventions should prioritize building strong social networks and support systems for individuals with mental disorders. Enhancing social skills and promoting positive organizational cultures in healthcare settings can significantly improve the effectiveness of peer support programs. To tackle this challenge, well-designed training programs for consumers, managers, and healthcare professionals are essential to enhance their understanding of the pivotal role played by peer support.

### 4.4. Gazing at the Desired Horizons

The participants, acting as peer support workers, expressed a strong desire to transcend their current circumstances. Despite facing recurrent relapses and isolation, they remained determined to effect positive change. Their aspirations included leading ordinary lives, contributing to the advancement of mental health, and combating societal prejudice and stigma. These goals aimed at legitimizing peer support work as an established profession and fostering solidarity among those with mental health issues, underscoring the importance of structured training and activities in peer support [36,37]. Research has highlighted various obstacles encountered by peer support workers (PSWs) within the recovery-oriented practice (ROP) framework, such as insufficient support from institutions, imbalanced power dynamics, negative staff attitudes, and a lack of professional recognition [38]. These issues often lead to significant challenges and constraints for PSWs in their professional roles. To enhance their contributions effectively, a combination of institutional support and acknowledgement of their competencies as colleagues is crucial, along with creating a supportive and inclusive environment. From the viewpoint of mental health professionals, acknowledging and aiding the resolve of individuals facing mental health challenges to surmount difficulties and make meaningful contributions to their lives and the wider mental health community is critical [39]. Promoting the establishment of peer support as a recognized profession could yield substantial benefits for both individuals with mental health conditions and the community at large.

### 4.5. Awakening the Inner Hero

The study identified “Awakening the Inner Hero” as a key theme among participants in peer support activities. Individuals with mental illnesses, often subjected to societal stereotypes of passivity and weakness [40], experience a transformative journey through these activities. Engaging in peer support roles enables them to shift from passive bystanders to active protagonists in their recovery, boosting their self-confidence and sense of purpose. Prior research supports the positive influence of peer support in empowering these individuals, facilitating their recovery, and promoting stable employment [41]. Furthermore, recognizing the potential and strengths of individuals in peer support roles is essential [42]. This aligns with expert perspectives on the importance of self-acceptance and active participation in one’s recovery journey, contributing significantly to overall well-being [43]. In essence, peer support activities not only challenge societal stereotypes but also empower individuals with mental disorders to be the architects of their own recovery.

### 4.6. Standing as a Person Who Cherishes Life

In the final theme, participants engaged in peer support activities and embraced the concept of “Standing as a Person Who Cherishes Life”. This transformation allowed them to discover new aspects of their humanity, experience the joy of learning, and gain economic assistance. These factors collectively contributed to their personal growth and stability, even in the presence of ongoing symptoms. Notably, they achieved a level of independence. In Bart et al.’s study [44], an investigation was conducted into the perceptions of both nurses and peer support workers regarding their respective roles in mental health practice. It revealed that peer support workers effectively harnessed their firsthand experiences, emphasizing identity construction, empowerment, and competence development in nursing education and training. The participants in our study also derived tangible benefits, including earning wages, improving interpersonal relationships, and engaging in leisure activities, through their involvement in peer support activities. This underscores the significance of fostering self-acceptance and highlights the imperative role of psychiatric nurses in facilitating and supporting the expansion of peer support activities. From a mental health professional perspective, it is vital to actively promote and endorse the expansion of peer support activities as an integral component of comprehensive mental health care. This collaborative approach significantly contributes to individuals’ overall well-being, enriches their recovery process, and ultimately leads to improved mental health outcomes, even in the presence of symptoms.

### 4.7. Limitations

Two notable limitations warrant acknowledgment within this study. Firstly, participant selection may have been influenced by variables such as geographical location, potentially introducing bias into the study’s outcomes. Secondly, the unforeseen impact of the pandemic situation led to difficulties in perceiving facial expressions during in-person interviews due to the use of masks. Additionally, this situation might have imposed limitations on participants’ interactions with the interviewers.

## 5. Conclusions

This study’s findings highlight the multifaceted impact of peer support activities, underscoring a transformative journey for participants. This process involves confronting personal challenges, fostering meaningful relationships, aspiring for positive changes, and awakening inner strengths. Significantly, this transformation includes embracing life even amid persistent symptoms, illustrating the resilience-oriented practice perspective. ROP focuses on strengths and resilience rather than deficits, a view clearly mirrored in these peer support activities. Participants not only cope with their mental health challenges but also discover and leverage their resilience, contributing positively to their own and others’ lives. The study emphasizes the critical role of peer support in holistic mental health care and suggests that expanding such activities could greatly enhance well-being and improve mental health outcomes.

Advocating for a broader recognition and integration of peer support, this research aligns with the ROP approach by emphasizing the importance of resilience and personal growth in mental health services. Integrating peer support as a core component in mental health services not only enhances the recovery process but also fosters a more inclusive and empathetic society.

## Figures and Tables

**Table 1 healthcare-12-00123-t001:** Participant information.

Characteristics	Participant 1	Participant 2	Participant 3	Participant 4	Participant 5
Gender	Female	Female	Male	Female	Female
Age	59	54	50	54	40
Marital Status	Married	Divorced	Divorced	Married	Married
Income Type	Irregular	Regular	Regular	Irregular	Irregular
Education Level	High School	High School	Withdrawal College	Bachelor’s Degree	Associate’s Degree
Diagnosis	Bipolar Disorder	Schizophrenia	Schizophrenia	Schizophrenia	Bipolar Disorder
Disability Registration	Yes	Yes	No	No	Yes
Illness Duration	25	24	26	27	13
Peer Support Worker Duration	5	4	4	3	4

**Table 2 healthcare-12-00123-t002:** Example of the First Three Steps of Interpretative Phenomenological Analysis.

1st Step Original Transcript	2nd Step Example of Initial Noting (Exploratory Commentary)	3rd Step Generated Initial Theme
“Change is about how I wasn’t able to have long conversations with others. But now, after living like this, I can confidently speak to others. And I have a few friends who call me for help. So, I’m giving counseling, though not officially… These processes have indeed led to another recovery for me, as I am in the stage of recovery. Have I come to know more people? It’s that kind of time.”	- Does knowing more people mean something? What’s the meaning or significance? - Use of words like help, conversation, counseling. - Change in the participant’s situation to listening to others’ stories and providing help? (changes in temporal and relational situations)	Becoming a source of help and support for others, reflecting personal growth and increased social connections.

**Table 3 healthcare-12-00123-t003:** Examples of the Fourth Step of Interpretative Phenomenological Analysis.

Example of ‘Polarization Strategy’		
Challenges Faced in Peer Support Activities	Benefits Gained from Peer Support Activities	
- Compulsion to know a lot and work hard for the activities.	- Used to avoid and run away, but now bearing and enduring.	
- Feeling ignored and conscious of the supervisors at the workplace.	- Reflecting on and uplifting myself.	
- Difficult to quit despite hardships because of others’ presence.	- Gaining a support system and helpful aspects in life (financial, interpersonal).	
Example of ‘Contextualization Strategy’		
Contextual Theme	Page/Line	Keywords
Enduring Challenges for Growth		
- Challenges and growth process as an employee in peer support.	4. 18	Supervisor, harsh words, criticism.
- Enduring challenges to bloom beautifully.	5. 11	Beautiful flower, endure.
- Will to protect oneself and attention from others.	10. 9	Watchful eyes, disappointment.
- Enduring situations without self-blame or avoidance in peer support.	17. 13	Avoiding, enduring.
Example of ‘Integration Strategy’		
Context	Integration Theme	
Deciding to engage in peer support activities	- Starting activities with the thought of aiding recovery and helping others.	
- Initial compulsion to work hard for the activities.		
Experiencing challenges in peer support	- Feeling tense and watched by supervisors while participating in peer support employment.	
- Hearing harsh words and receiving criticism.		
Realizations from peer support activities	- Gaining a sense of leading and creating one’s life.	
- Viewing experiences in peer support as typical of any employee.		
- Developing resilience and steadfastness.		

**Table 4 healthcare-12-00123-t004:** Examples of the Fifth and Sixth Steps of Interpretative Phenomenological Analysis.

Identifying Patterns from Cases		Themes and Subthemes
Focus on Relationships		Navigating the Paths of Relationships
Recognized and supported by family	Participant 4	Discovering relationship significance through peers’ experiences
Becoming a trusted person	Participant 3	
Acknowledged by family	Participant 1	
Supported by family	Participant 5	
Respected by others	Participant 2	Connecting with others through active listening
Connecting and listening to others	Participant 4	
Changing to help and listen to others	Participant 2	
Focus on Self		Rising and Refining Myself
Revealing oneself	Participant 1	Unveiling and embracing one’s true self
Developing self-esteem	Participant 5	
Becoming brighter and more confident	Participant 1	
Becoming indispensable to others	Participant 2	Developing inner strength to overcome challenges
Gaining pride and achievement	Participant 3	
Finding hidden advantages	Participant 4	
Discovering inner strengths	Participant 1	
Gaining confidence	Participant 4	

**Table 5 healthcare-12-00123-t005:** Themes and Subthemes.

Themes	Subthemes
Facing confusion and challenges	Aspiring and striving due to a desire to excel
	Feeling drained and encumbered by symptoms
Rising and refining myself	Unveiling and embracing one’s true self
	Developing inner strength to overcome challenges
Navigating the paths of relationships	Discovering relationship significance through peers’ experiences
	Connecting with others through active listening
Gazing at the desired horizons	Cultivating hope in adverse circumstances
	Working towards a vision of the future and making efforts
Awakening the inner hero	Finding the driving force and meaning in one’s life
	Facing challenges without fear
Standing as a person who cherishes life	Growing and evolving without lingering on symptoms
	Advocating for rights in safe spaces

## Data Availability

The datasets utilized in this study can be made available upon reasonable request from the corresponding author. The data are not publicly available due to ethical reasons.

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
