# Peer review of "The Poetry of Recovery in Peer Support Workers with Mental Illness: An Interpretative Phenomenological Analysis"

_healthcare, 2024, doi:10.3390/healthcare12020123_

Round 1
Reviewer 1 Report
Comments and Suggestions for Authors
Peer reviewed article:
The Poetry of Recovery in Peer Support Workers with Mental Illness: An Interpretive Phenomenological Analysis
Thank you for the opportunity to review your insightful paper. Any work about lived experience peer workers is greatly needed in the limited literature internationally.
I thoroughly enjoyed reading your article. I could find no issues with the paper methodologically. It was thorough and the article was easily read.
There wasn’t much about recovery-oriented practice (ROP) explained in your paper. I would like you to elucidate on that a bit more. You touch on it but do not make it explicit enough. Your paper is clearly about ROP and the introduction of peer work. There is plenty of literature on this if you look. Perhaps you did not think some of it was relevant but I believe you could find some valuable insights into other work being done in this space.
There was no real research question available so I would request you target a question that is suitable for what you hoped to get out of your research.
You mention South Korea in the abstract but do not mention it anywhere else in the paper. It would be good for the reader to be reminded the location of the research.
It addresses a gap in the field of peer workers in mental health but you don’t emphasise this enough. There is little research on peer workers in the recovery-oriented space (ROP, so melding these two point together would benefit the article. Recovery-oriented practice is taking over the western countries in different forms and specifying the type of ROP you are focusing on would be beneficial.
In the methodology I would have liked to see more peer worker quotes analysed by content analysis if there weren’t many or thematic analysis if there were more. This would strengthen you paper’s methods and results. You could have used a table or graph to show some research questions and responses.
You mention psychiatric nurses in the conclusion but not much anywhere else in you research. The conclusion did not show that a specific question was responded to, and whether any specific question was answered. You need to strengthen this by having a question you are focused on, and answering it in your results. I really want more papers such as yours published in the peer worker space in mental health but you do need to make it stronger.
I wonder , however about your reading of the literature that is out there on lived experience peer workers that you overlooked. I would like to see a bit more of the great work that is out there on this very topic.
For example just a few:
Chisholm, J & Petrakis, M. (2020) and (2021)
Clossey et al., (2018)
Kemp, V., & Henderson A. (2012). A bit old but worth a read.
O’Connor, N., Clark, S., & Ryan C.J. (2017)
Vandewalle et al. (2016)
Byrne, L., Happell, B., & Reid-Searl, K (2016)
Anything by Louise Byrne an Australian reseracher
Gallgher, C. (2014) Another old one but good
Anything by Brenda Happell and Australian researcher, although mainly on lived experience in education.
Roennfeldt, H. Byrne, L. (2020)
Many of these are Australian researchers. Australia, especially Victoria is doing a lot in the ROP and lived experience peer worker space.
Again than you for the opportunity to review your paper. I believe just a tad more explanation of ROP and citing some other lived experience peer worker articles could make it better, and sit in the area of ROP and peer work more broadly.
Author Response
Letter to the Reviewer:
Dear Reviewer,
I trust this note reaches you in excellent health and high spirits. I am compelled to express my deep appreciation for the comprehensive and insightful critique you provided on my work, "The Poetry of Recovery in Peer Support Workers with Mental Illness: An Interpretive Phenomenological Analysis."
It is indeed encouraging to learn that you regard the methodology of my study as robust and find the article to be reader-friendly. Such positive feedback reinforces my dedication to upholding the quality of my research endeavors.
Your valuable recommendation to expand on Recovery-Oriented Practice (ROP) in the paper has opened new avenues for me. I acknowledge the need for a deeper and more explicit examination of this topic to augment the paper’s significance and depth. Presently, I am exploring further literature on this subject to enhance my depiction and understanding of ROP in my study's framework.
Moreover, I am particularly thankful for your expert guidance and the specific points you highlighted for improvement. Your insights have been vital in sharpening my arguments and elevating the overall impact of my work.
Being a researcher who aims to contribute valuable insights in our field, having the opportunity to have my work reviewed by esteemed professionals such as yourself is a great honor. Your encouraging and enlightening feedback has spurred me on, and I am keenly looking forward to incorporating your valuable suggestions into my manuscript.
In closing, your time, effort, and expertise devoted to reviewing my article are deeply appreciated. Your contributions have significantly enriched my academic journey and writing process.
Thank you once again for your invaluable support and guidance.
Respectfully,
Corresponding Author

Reviewer 2 Report
Comments and Suggestions for Authors
The authors have provided a clear description of the study methodology in section 2.3 "Procedure for Data Collection". Minor enhancements to this section, such as including average interview duration and providing more detail on the preliminary review of interview questions, could further strengthen the methodological rigor. Additionally, when presenting participant demographics in Table 1, specifying the regional context as "G Metropolitan City, South Korea" in the column header would help contextualize the data. Overall, the study is well-designed and findings offer valuable insights for mental health recovery. With a few refinements, the manuscript demonstrates strong potential for publication.
Comments on the Quality of English Language
The English language used in the manuscript is clear and easy to understand. Minor language edits could help enhance readability in some areas. For example, breaking lengthy paragraphs into shorter sections using subheadings where appropriate. Additionally, careful proofreading is recommended to correct typographical and grammatical errors. However, these issues do not significantly impact comprehension. On the whole, the quality of English is suitable for academic publication.
Author Response
Letter to the Reviewer:
.
Dear Reviewer,
I trust this note reaches you in excellent health and high spirits. I am compelled to express my deep appreciation for the comprehensive and insightful critique you provided on my work, "The Poetry of Recovery in Peer Support Workers with Mental Illness: An Interpretive Phenomenological Analysis."
It is indeed encouraging to learn that you regard the methodology of my study as robust and find the article to be reader-friendly. Such positive feedback reinforces my dedication to upholding the quality of my research endeavors.
I am particularly thankful for your expert guidance and the specific points you highlighted for improvement. Your insights have been vital in sharpening my arguments and elevating the overall impact of my work.
Being a researcher who aims to contribute valuable insights in our field, having the opportunity to have my work reviewed by esteemed professionals such as yourself is a great honor. Your encouraging and enlightening feedback has spurred me on, and I am keenly looking forward to incorporating your valuable suggestions into my manuscript.
In closing, your time, effort, and expertise devoted to reviewing my article are deeply appreciated. Your contributions have significantly enriched my academic journey and writing process.
Thank you once again for your invaluable support and guidance.
Respectfully,
Corresponding Author

Reviewer 3 Report
Comments and Suggestions for Authors
Dear authors, thank you for the manuscript. Please find below my concerns
1. The introduction should focus more on the current understanding of the issues faced by peer supporters with mental or other physical illness and how it may impact the clinical outcomes of the people to whom they provide service.
2. Paragraph 2 and 3 can create confusion for the readers. So, I would suggest removing it.
3. Line 93 a in "approach" is highlighted.
4. There is a need to elaborate the IPA method (in addition to lines 80-85) either in the introduction or methods explaining the advantage of this methdology.
5. Need to explain sample size (n=5)
6. Section 2.5. Need to provide information on whether participants were reimbursed of their time to the study.
7. The number of sub-themes for each theme was 2. Was this just by chance or planned?
8. Please check to the journal guidelines on whether participant quotes need to be provided as italic font.
9. Line 331 - what does 30 reflect to? It does not fit with the text.
10. It must be explained (in details) how the authors decided the theme labels
11. In the conclusion, the authors detail about psychological capital among the pyschiatric nurses. However, in the preceeding section, it was no where mentioned of psychiatric nurses. Can you please explain how conclusion explains your research. It looks like conclusion section was copy pasted.
Comments on the Quality of English Language
N/A
Author Response

(The authors gave the same response as above.)

Round 2
Reviewer 1 Report
Comments and Suggestions for Authors
Re-reviewed paper: The Poetry of Recovery in Peer Support Workers with Mental Illness: An Interpretive Phenomenological Analysis
Well done on your impressive efforts to restructure the article and address its challenges.
I believe you have addressed all of my concerns with the paper, and this has made it a much more clear article. The methodology has been greatly improved. I feel I could now replicate your study with ease. The theory is sound.
Thank you for you obvious extra reading of recovery-oriented practice literature. This has situated you article in the recovery space with much more significance and relatability to ROP.
You have cited many extra authors who are leaders in the field, which elevates your reference list and lifts it up to sit where it now deserves.
I thought the addition of quotes from participants was important to elevate the voices of participants but also to adds to the importance of the results section.
There was a “77” on page 12, line 447 that you will see when you proof read if your paper is accepted for publication.
I hope it is published as the mor literature elevating the voices and experiences of lived experience peer workers is vital in this changing mental health recovery area, ROP.
Best wishes.